# Purity Control Based on a Type-II Fuzzy Controller for a Simulated Moving Bed

**Chao-Fan Xie** [1] **and Rey-Chue Hwang** [2,*]

1    School of Big Data and Artificial Intelligence, Fujian Polytechnic Normal University, Fuqing 350300, China
2    Department of Electrical Engineering, I-Shou University, Kaohsiung 84001, Taiwan
*    Correspondence: rchwang@isu.edu; Tel.: +886-09-2872-2593

**Abstract:** The control of a simulated moving bed (SMB) is always a challenging chemical control topic due to its complexity and nonlinearity. Its mathematical model must undergo an affine transformation and digitization before it can be controlled. Basically, there are three aspects that need to be considered in the nonlinear control of an SMB. First, the nonlinear characteristics are more complicated due to the switching time parameters of discrete events. Second, the control objective is not to minimize the control output error, but to make the separated concentrations between the components of the substance reach a certain ratio. Finally, the control variables are highly coupled. So far, the vast majority of the industry still uses relatively simple PLC controls; a few use specific controllers based on materials to be separated such as model predictive controls and PID controllers. Therefore, there is no unified intelligent processing mode. In this paper, a type-II fuzzy controller is presented and used as an SMB control. The interference of the related parameters was tested to observe the stability and robustness of the controller. The type-II fuzzy control was based on type-II fuzzy sets, which resulted in the type-II fuzzy controller having more flexible attribution function values. The results showed that the type-II fuzzy controller was not only more accurate in the control, but also better for robustness and adaptability than an ordinary fuzzy controller and PID controller.

**Keywords:** simulated moving bed; type-II fuzzy controller; PID controller

## 1. Introduction

Chromatography is a technique that uses the distribution differences of a compound in the stationary and mobile phases to achieve the separation of a mixture. SMBs have been widely used in the manufacturing process of chemical and biomedical products and identified as a technology that can greatly improve the efficiency of chromatographic separation. However, due to the many complex parameters involved in the SMB control process, the cost of obtaining optimized parameters through actual experiments to control the separation process of an SMB is huge. Technicians often set a few initial control parameters of an SMB according to their experience or by experiments, but such a method often makes the control of an SMB unable to achieve its best state [1–3].

Many researchers hope to conduct quantitative analyses by studying the mathematical models of SMBs. Generally, there are two mathematical models adopted in an SMB; namely, the general rate model and the equilibrium diffusion model. The general rate model is a comprehensive mechanism model, which is more consistent with the actual process but more complicated. The equilibrium diffusion model is a simplified mechanism model that can reflect the actual process well when the concentration of the separated components is low [4]. However, no matter which model is used, they are a set of partial differential equations and the analytical solutions are difficult to obtain. It might even be said that the real solutions are impossible to obtain.

Traditional nonlinear theory and control methods can handle nonlinear autonomous systems, affine control systems, continuous dynamic differential equations, or difference

equations but cannot cope with the complexity of an SMB [5,6]. An SMB is different from the above systems. It is a nonaffine and nonautonomous system; the most difficult part of an SMB control is that the discrete switching time variable needs to be considered for the actual system. Due to the existence of this variable, the continuous dynamic behavior of the system is destroyed and the SMB becomes a new type of hybrid system. In the current methods, there is no effective solution for controlling such a hybrid system; thus, how to control an SMB efficiently and precisely is still a challenging research topic [7,8].

In past research, Nogueira et al. proposed a work about the control of an SMB process in the enantio-separation of binaphthol [9]. A switching system-based strategy and a proper transfer function were used in the stable model predictive controller (MPC) to overcome the problems related to the dynamic behavior of the process. Yan et al. applied two subspace system identification algorithms—a multivariable output error state-space (MOESP) identification algorithm and a numerical algorithm—to obtain the third-order and fourth-order state-space yield models of an SMB, respectively. The MPC method based on the established state-space model was then used in an SMB chromatographic separation process [10]. Other related SMB control studies include a self-tuning control and an adaptive nonlinear model predictive control [11,12]. Marı'a-Sonia et al. considered the general dynamic optimization (open loop optimal control) of an SMB chromatographic separation process. This allowed the calculation of an optimal feed concentration and/or feed flow rate over each switching period with maximum flexibility [13]. In a practical application, Coelho et al. used the mathematical optimization method to predict the performance of a real SMB system and achieved a 97% separation purity control of tartronic acid (TTA) and glycemic acid (GCA) materials [14]. To optimize the performance of the control system, Li et al. proposed a predictive control method of the SMB chromatographic separation process based on a piecewise affine model [15]. A bypass-simulated moving bed (BP-SMB) chromatographic process characterized by the possibility of over-purifying the raffinate and/or the extract product followed by blending with the feed; this was studied by Maruyama et al. [16].

Other studies using mixed multiple models are described below. Suvarov et al. used a self-adjusting control to adjust the spatial position of adsorption and desorption waves and then adjusted the purity and productivity of the raffinate and extraction flow. Such a predictive control technology has been widely used in program controls [17]. Song et al. put forward a new operation strategy called Simcon. This method improved the separation performance of SMB chromatography by simultaneously controlling the outlet and inlet. This control operation was simple, but the accuracy was not too high [18]. Carols et al. proposed a new approach that was based on the combination of wave theory and a multimodel predictive control (MMPC). The wave theory provided the theoretical framework [19]. Based on the mathematical model, a numerical solution process was proposed to simulate the transient and steady state of a moving bed by Leao et al. It was also a theoretical architecture and computing model [20]. In a simulation study, Ju WenLee proposed a simplified process model with linear isotherms to estimate the process state of SMB chromatography. It obtained optimal operating parameter conditions through a "switch by switch" switching operation within the moderate nonlinear range of the Langmuir isotherm [21]. Yang et al. proposed an optimization strategy based on an improved moving asymptote algorithm. Research has shown that an optimal controller based on an improved moving asymptote method can dynamically control and optimize the process of a simulated moving bed [22].

Generally, these studies were for specific equipment and separation materials and not a general solution. For example, sugar separation in the food industry may not be applicable to the chemical industry. In terms of the calculation model, most are based on the finite element calculation method. Although the precision of a finite element calculation is high, the calculation efficiency is not high enough; therefore, it is not suitable for the application of a real-time online control. In this paper, the authors attempted to use a finite difference method to enhance the calculation efficiency. The robustness and

adaptability of the controller used had a lack of corresponding research. With a change in the environmental parameters, the results could easily to lead to the failure of the separation effect. In the practical application of an SMB, it is often necessary to adjust it according to changes to the equipment and in the environment; thus, a more intelligent automatic control technology is needed. Compared with traditional nonlinear control methods such as feedback linearization, synovium, and pole placement, a fuzzy control based on fuzzy mathematics does not need extensive mechanism model knowledge; thus, the use of fuzzy control technology is extremely convenient and effective in mathematical processing, especially for nonlinear systems with complex structures.

In the face of highly sensitive SMB nonlinear systems, the simple use of fuzzy control technology cannot guarantee the robustness and adaptability of its control. The fuzzy controller depends on fuzzy application rules as well as the force size and other parameters. Similarly, if the controller parameters are not properly selected, it is easy to cause it to cross the feasible separation area of the SMB system in the control, resulting in pathological characteristics. In this paper, the authors propose a more effective type-II fuzzy controller. A type-II fuzzy control is based on type-II fuzzy sets, which provides the type-II fuzzy controller with more flexible attribution function values because the value of the type-II fuzzy function is also a fuzzy set. Thus, it makes the design of the type-II fuzzy inference more flexible; the error tolerance of the system also increases. Furthermore, the robustness and adaptability of the system control are also enhanced. The rest of the paper is organized as follows. Section 2 presents the mathematical model of the SMB. In Section 3, the Crank–Nicolson method used to numerate the PDEs is presented. Section 4 describes the setting of the simulation parameters. In Section 5, the type-II fuzzy controllers are designed to apply to the SMB system. Section 6 shows the experimental results. Section 7 is the conclusion.

## 2. SMB Mathematical Model

For an SMB, the balance between the mobile and solid phase is given by:

$$\frac{\partial C_{ij}}{\partial t} = D_i \frac{\partial^2 C_{ij}}{\partial x^2} - v_j^* \frac{\partial C_{ij}}{\partial x} - \frac{1-\varepsilon}{\varepsilon} k_i (q_{ij}^* - q_{ij}) \tag{1}$$

$$\frac{\partial q_{ij}}{\partial t} = k_i (q_{ij}^* - q_{ij}) \tag{2}$$

The meaning of the variables is shown in Table 1.

**Table 1.** Meanings of the variables of the SMB system.

| Parameter | Nomenclature |
|---|---|
| $x\,(\text{cm})$ | Axial distance |
| $L\,(\text{cm})$ | Column length |
| $d\,(\text{cm})$ | Column diameter |
| $k_A\,(\text{gL}^{-1})$ | Comprehensive mass transfer constant of $A$ |
| $k_B\,(\text{gL}^{-1})$ | Comprehensive mass transfer constant of $B$ |
| $H$ | Henry constant |
| $v^*\,(\text{cm min}^{-1})$ | Effect velocity of body |
| $u_s\,(\text{cm min}^{-1})$ | Solid flow rate |
| $C\,(\text{gL}^{-1})$ | Mobile phase concentration |
| $q\,(\text{gL}^{-1})$ | Solid phase concentration |
| $q^*\,(\text{gL}^{-1})$ | Solid phase concentration at the equilibrium between the solid phase and mobile phase |
| $Q\,(\text{cm}^3\,\text{min}^{-1})$ | Volume flow rate |
| $T$ | Switch time |
| $D\,(\text{cm}^2\,\text{min}^{-1})$ | Effective dispersion coefficient |
| $\varepsilon$ | Bulk void fraction |
| $i$ | Material index: $A$ or $B$ |
| $j$ | Column number: 1, 2, 3, 4, 5, 6, 7, or 8 |

From Equations (1) and (2), we obtained:

$$\frac{\partial C_{ij}}{\partial t} = D_i \frac{\partial^2 C_{ij}}{\partial x^2} - v_j^* \frac{\partial C_{ij}}{\partial x} - \frac{1-\varepsilon}{\varepsilon} \frac{\partial q_{ij}}{\partial t} \tag{3}$$

The adsorption equilibrium of the two materials could be expressed by a linear isotherm:

$$q_{ij} = H_i C_{ij} \tag{4}$$

The purity formulas could be defined by:

$$\bar{C}_{E,B} = \frac{C_{E,B}}{C_{E,A} + C_{E,B}} \tag{5}$$

$$\bar{C}_{R,A} = \frac{C_{R,A}}{C_{R,A} + C_{R,B}} \tag{6}$$

$\bar{C}_{E,B}$ represents the purity of material $B$ from the extraction outlet, $\bar{C}_{R,A}$ represents the purity of material $A$ from the raffinate outlet, represents the concentration of material $B$ from the extraction outlet, and $C_{E,B}$ represents the concentration of material $A$ from the raffinate outlet [23,24].

The fixed-value conditions of the SMB model were as follows:

(a) Initialcondition : $t = 0$ $\qquad\qquad\qquad\qquad\qquad$ $C_{ij}(x,0) = 0$ $\tag{7}$

(b) Endofcolumn : $\qquad\qquad\qquad\qquad\qquad$ $\frac{\partial C_{ij}(x,t)}{\partial x}\Big|_{x=L_{end}} = 0$ $\tag{8}$

(c) Headofcolumn : $\quad D_i \frac{\partial C_{ij}(x,t)}{\partial x}\Big|_{x=L_0} = v_j^*[C_{ij}(L_0,t) - \bar{C}_{ij}^{\,sec\,t}(t)]$ $\tag{9}$

$x = L_{end}$ and $x = L_0$ represented the boundaries of the fixed-value condition; that is, the conditions that needed to be met at the end and initial position of the column. $\bar{C}_{ij}^{\,sec\,t}(t)$ was related to the zone where it was located.

## 3. Using the Crank–Nicolson Method to Numerate the PDEs

The Crank–Nicolson method was then used to establish the discrete dynamic system of the simulated moving bed so that the controller could be used as a real-time control.

We set $t_k = t_0 + ks$, $x_l = x_0 + lh$, $F = \frac{1-\varepsilon}{\varepsilon}$. It can get the equations as follow:

$$\frac{\partial^2 C_{i,j}}{\partial x^2}(x_l, t_k) = \frac{C_{i,j}(x_{l+1}, t_k) - 2C_{i,j}(x_l, t_k) + C_{i,j}(x_{l-1}, t_k)}{h^2} \tag{10}$$

$$\frac{\partial C_{i,j}}{\partial x}(x_l, t_k) = \frac{C_{i,j}(x_{l+1}, t_k) - C_{i,j}(x_l, t_k)}{h} \tag{11}$$

$$\frac{\partial C_{i,j}}{\partial t}(x_l, t_k) = \frac{C_{i,j}(x_l, t_{k+1}) - C_{i,j}(x_l, t_k)}{s} \tag{12}$$

$$\frac{\partial q_{i,j}}{\partial t}(x_l, t_k) = H_i \frac{\partial C_{i,j}}{\partial t}(x_l, t_k)$$
$$i = 1, 2, j = 1, \cdots 8 \tag{13}$$

$C_{i,j}(x_l, t_k)$, denoted as $C_{i,j}(l, k)$ and substituted into the SMB system, could simplify the formula symbols as follows:

$$(1 + FH_i)C_{i,j}(l, k+1) = \left(\frac{Ds}{h^2} - \frac{v_j^* s}{h}\right)C_{i,j}(l+1, k) +$$
$$\left(\frac{v_j^* s}{h} - \frac{2Ds}{h^2} + 1 + FH_i\right)C_{i,j}(l, k) - \frac{Ds}{h^2}C_{i,j}(l-1, k) \tag{14}$$

The initial conditions and boundary conditions also needed to be discretized as follows:

$$\text{Section 1 , first column : } \overline{C}_{i,j}{}^{\text{I}}(t) = \frac{Q_{IV}C_{i,j-1}(l_{n-1},t)}{Q_I} \tag{15}$$

$$\text{Section 3 , first column : } \overline{C}_{i,j}{}^{\text{III}}(t) = \frac{Q_{II}C_{i,j-1}(l_{n-1},t) + Q_f C_{f,i}}{Q_{III}} \tag{16}$$

$$\text{Any other column : } \overline{C}_{i,j}{}^{\sec t}(t) = C_{i,j-1}(l_{n-1},t) \tag{17}$$

$\overline{C}_{i,j}{}^{\sec t}$ was the column inlet concentration with superscript $\sec t = I, II, III, IV$ where $C_f$ was the feed stream concentration, $C_{0,i,j}$ was the initial concentration, and $l$ was the column length. From Equation (8), we obtained:

$$C_{i,j}(n+1,k) = C_{i,j}(n,k) \tag{18}$$

From Equation (9), we could then obtain:

$$D_i\left[\frac{C_{i,j}(1,k) - C_{i,j}(0,k)}{h}\right] = v_j{}^*[C_{i,j}(0,k) - \overline{C}_{i,j}{}^{\sec t}(k)] \tag{19}$$

We simplified it as:

$$C_{i,j}(1,k) = \left(\frac{hv_j{}^*}{D_i} + 1\right)C_{i,j}(0,k) - \frac{hv_j{}^*}{D_i}\overline{C}_{i,j}{}^{\sec t}(k) \tag{20}$$

We then set $M = \frac{v^*s}{h}, N = \frac{Ds}{h^2}$ to obtain the next two boundary equations:

$$\begin{aligned}
(1+FH_i)C_{i,j}(1,k+1) &= (1+FH_i+M-2N-\tfrac{MN}{M+N})C_{i,j}(1,k) \\
&+ (N-M)C_{i,j}(2,k) - \tfrac{M^2}{M+N}\overline{C}_{i,j}{}^{\sec t}(k)
\end{aligned} \tag{21}$$

$$\begin{aligned}
(1+FH_i)C_{i,j}(l,k+1) &= (N-M)C_{i,j}(l+1,k) + \\
(M-2N+1+FH_i)C_{i,j}(l,k) &- NC_{i,j}(l-1,k)l \neq 1, n
\end{aligned} \tag{22}$$

$$(1+FH_i)C_{i,j}(n,k+1) = (1+FH_i-N)C_{i,j}(n,k) - NC_{i,j}(n-1,k) \tag{23}$$

This denoted the matrix:

$$A = $$
$$\begin{bmatrix}
\frac{(1+FH_i+M-2N-\frac{MN}{M+N})}{(1+FH_i)} & \frac{N-M}{(1+FH_i)} & 0 & \cdots & 0 \\
\frac{-N}{(1+FH_i)} & \frac{(M-2N+1+FH_i)}{(1+FH_i)} & \frac{(N-M)}{(1+FH_i)} & \cdots & 0 \\
\vdots & \vdots & \vdots & \vdots & \vdots \\
0 & \cdots & \frac{-N}{(1+FH_i)} & \frac{(M-2N+1+FH_i)}{(1+FH_i)} & \frac{(N-M)}{(1+FH_i)} \\
0 & \cdots & 0 & \frac{-N}{(1+FH_i)} & \frac{(1+FH_i-N)}{(1+FH_i)}
\end{bmatrix}$$

$$w(k) = \begin{pmatrix} \frac{M^2}{M+N}\overline{C}_{i,j}{}^{\sec t}(k) & 0 & \cdots & 0 & 0 \end{pmatrix}^T$$

Finally, we obtained the iterative equation:

$$C_{i,j}(k+1) = AC_{i,j}(k) + w(k) \tag{24}$$

## 4. Simulation Settings

In order to observe the operation of the SMB system, a 2-2-2-2 SMB model with 4 zones, as shown in Figure 1, was implemented. Each zone had two columns. The initial settings

of all SMB parameters are shown in Table 2. In the simulation, the relationship between the flow rate, the velocity of the body $v_j^*$, and the volume flow rate $Q_j$ was expressed as Equation (15), where $r$ was the radius of the column and $\varepsilon$ was the bulk void fraction. The operational time step was set to be 0.1 s and the length of the string in the space was divided into 100 parts.

$$v_j^* = \frac{Q_j}{\varepsilon \pi r^2} \tag{25}$$

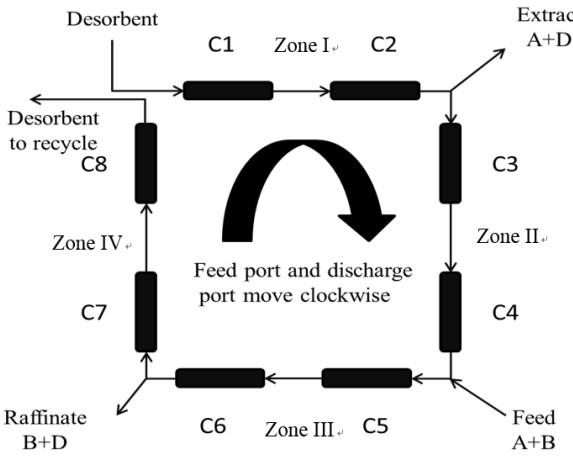

**Figure 1.** SMB model with four zones.

**Table 2.** The initial parameters for the separation.

| Parameter | Value | Parameter | Value |
|---|---|---|---|
| $L(\text{cm})$ | 25 | $C_{f,i}(\text{gL}^{-1})$ | 5 |
| $d(\text{cm})$ | 0.46 | $\theta(\text{min})$ | 3 |
| $H_A$ | 0.001 | $Q_I(\text{cm}^3\text{min}^{-1})$ | 6.75 |
| $H_B$ | 0.45 | $Q_{II}(\text{cm}^3\text{min}^{-1})$ | 6.6 |
| $D_A(\text{cm}^2\text{min}^{-1})$ | 0.2 | $Q_{III}(\text{cm}^3\text{min}^{-1})$ | 7 |
| $D_B(\text{cm}^2\text{min}^{-1})$ | 1.265 | $Q_{IV}(\text{cm}^3\text{min}^{-1})$ | 2 |
| Spatial number | 50 | $\varepsilon$ | 0.8 |

All calculations were conducted in MATLAB R2016a on a PC equipped with an Intel core i7-3770K with 3.53 GHz, 16 GB RAM, and running Windows 10.

## 5. Type-II Fuzzy Controller Design

In this research, a type-II fuzzy controller was designed to control the SMB system. A traditional fuzzy controller was also used as a comparison. In the fuzzy mechanism, the purity error ($e(k)$) and error change ($\Delta e(k)$) of materials B and A were used as the fuzzy inputs and defined as follows ($C_{E,B}$ and $C_{R,A}$ were the immediate concentrations of B and A sensed at the extraction and raffinate outlets, respectively):

$$e_1(k) = desired\ B - C_{E,B} \tag{26}$$

$$e_2(k) = desired\ A - C_{R,A} \tag{27}$$

$$e_3(k) = e_1(k) + e_2(k) \tag{28}$$

$$\Delta e_1(k) = e_1(k) - e_1(k-1) \tag{29}$$

$$\Delta e_2(k) = e_2(k) - e_2(k-1) \tag{30}$$

$$\Delta e_3(k) = e_3(k) - e_3(k-1) \tag{31}$$

where $e_1(k)$ and $\Delta e_1(k)$ were the inputs of the zone I controller, $e_2(k)$ and $\Delta e_2(k)$ were the inputs of the zone II controller, and $e_3(k)$ and $\Delta e_3(k)$ were the inputs of the zone III controller.

In the control process of the SMB, the flow rates of zone I ($Q_1$), zone II ($Q_2$), and zone III ($Q_3$) were controlled by three independent fuzzy controllers. The common structure of the fuzzy control process is shown in Figure 2.

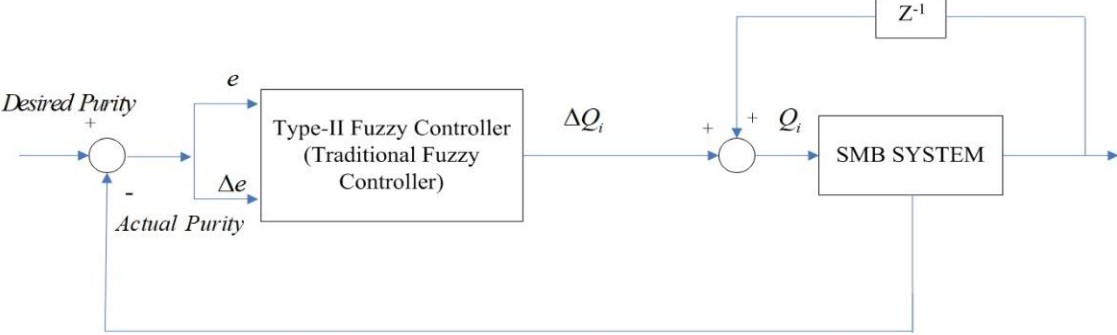

**Figure 2.** The structure of the SMB control process.

For the traditional fuzzy controller, five linguistic variables—namely, NB, NS, ZE, PS, and PB—were defined for both $e_i(k)$ and $\Delta e_i(k)$. NB represented a large negative force, NS represented a medium negative force, ZE represented no force, PS represented a medium positive force, and PB represented a large positive force. The triangular membership function and singleton membership function shown in Figures 3 and 4 were used as the fuzzifier and defuzzifier, respectively. The rule table for the three controllers is listed in Table 3. The singleton values $[a_1, a_2, a_3, a_4, a_5]$ set for $\Delta Q_i$ were $\Delta Q_1 = [-0.12, -0.08, 0, 0.08, 0.12]$, $\Delta Q_2 = [-0.006, 0.004, 0, 0.004, 0.006]$, and $\Delta Q_3 = [-0.08, -0.05\ 0, 0.05, 0.008]$.

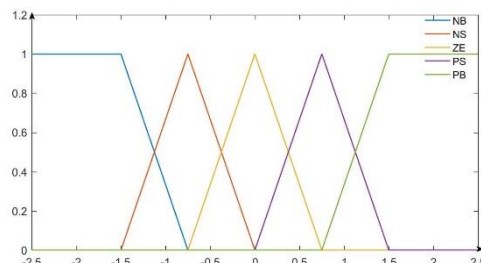

**Figure 3.** The fuzzifier membership function of the traditional fuzzy controller ($e_i(k)$ and $\Delta e_i(k)$).

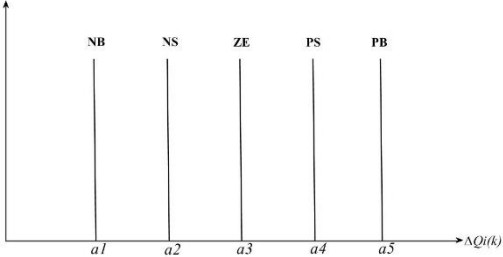

**Figure 4.** The defuzzifier membership function of the traditional fuzzy controller.

**Table 3.** The rule table for the traditional fuzzy controller for $\Delta Q_i$ ($i$ = 1, 2, 3).

| $\Delta e(k)$ \ $e(k)$ | NB | NS | ZE | PS | PB |
|---|---|---|---|---|---|
| NB | NB | NB | NB | NB | NS |
| NS | NB | NB | NB | NS | ZE |
| ZE | NB | NS | ZE | PS | PB |
| PS | NB | NS | PS | PB | PB |
| PB | NS | ZE | PB | PB | PB |

In the type-II fuzzy system, three linguistic variables—namely, NS, ZE, and PS—were defined for both $e_i(k)$ and $\Delta e_i(k)$ ($i$ = 1, 2, 3). The Gaussian membership function and singleton membership function shown in Figures 5 and 6 were used for the fuzzifier and defuzzifier, respectively. The rule table for the type-II controller is listed in Table 4. The singleton values ($c_1$, $c_2$, $c_3$) set for $\Delta Q_i$ were $\Delta Q_1$ = [−0.1, 0, 0.1], $\Delta Q_2$ = [−0.1, 0, 0.1], and $\Delta Q_3$ = [−0.6, 0, 0.6].

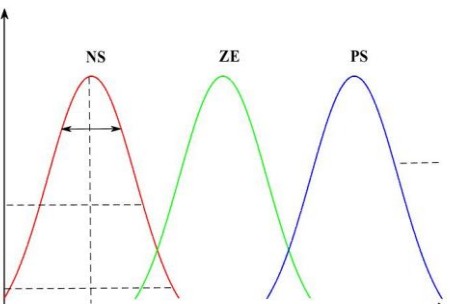

**Figure 5.** The fuzzifier membership function of the type-II controller ($e_i(k)$ and $\Delta e_i(k)$ ).

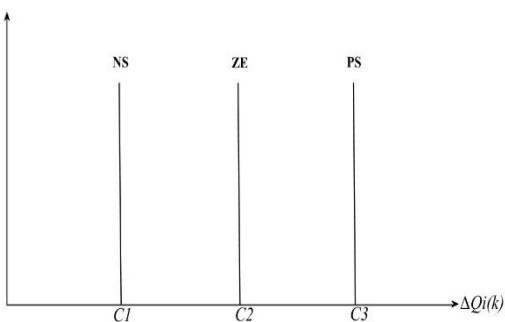

**Figure 6.** The defuzzifier membership function of the type-II controller.

**Table 4.** Type-II fuzzy rule table for ($i$ = 1, 2, 3).

| $\Delta e(k)$ \ $e(k)$ | NS | ZE | PS |
|---|---|---|---|
| NS | NS | NS | ZE |
| ZE | NS | ZE | PS |
| PS | ZE | PS | PS |

For the type-II fuzzy membership function, the center values and standard deviations were as follows:

$$
\begin{pmatrix}
m_{11} & \sigma1_{11} & \sigma2_{11} & m_{12} & \sigma1_{12} & \sigma2_{12} \\
m_{21} & \sigma1_{21} & \sigma2_{21} & m_{22} & \sigma1_{22} & \sigma2_{22} \\
m_{31} & \sigma1_{31} & \sigma2_{31} & m_{32} & \sigma1_{32} & \sigma2_{32} \\
m_{41} & \sigma1_{41} & \sigma2_{41} & m_{42} & \sigma1_{42} & \sigma2_{42} \\
m_{51} & \sigma1_{51} & \sigma2_{51} & m_{52} & \sigma1_{52} & \sigma2_{52} \\
m_{61} & \sigma1_{61} & \sigma2_{61} & m_{62} & \sigma1_{62} & \sigma2_{62} \\
m_{71} & \sigma1_{71} & \sigma2_{71} & m_{72} & \sigma1_{72} & \sigma2_{72} \\
m_{81} & \sigma1_{81} & \sigma2_{81} & m_{82} & \sigma1_{82} & \sigma2_{82} \\
m_{91} & \sigma1_{91} & \sigma2_{91} & m_{92} & \sigma1_{92} & \sigma2_{92}
\end{pmatrix}
=
\begin{pmatrix}
-0.25 & 0.4 & 0.0005 & -0.25 & 0.4 & 0.0005 \\
-0.25 & 0.4 & 0.0005 & 0 & 0.012 & 0.0008 \\
-0.25 & 0.4 & 0.0005 & 0.25 & 0.4 & 0.0005 \\
0 & 0.012 & 0.0008 & -0.25 & 0.4 & 0.0005 \\
0 & 0.012 & 0.0008 & 0 & 0.012 & 0.0008 \\
0 & 0.012 & 0.0008 & 0.25 & 0.4 & 0.0005 \\
0.25 & 0.4 & 0.0005 & -0.25 & 0.4 & 0.0005 \\
0.25 & 0.4 & 0.0005 & 0 & 0.012 & 0.0008 \\
0.25 & 0.4 & 0.0005 & 0.25 & 0.4 & 0.0005
\end{pmatrix}
$$

$m_{ij}$ represented the center value of the first-order membership function, $\sigma1_{ij}$ represented the standard deviation value of the first-order membership function, and $\sigma2_{ij}$ represented the standard deviation value of the second-order membership function where $i$ was the $ith$ rule and $j$ was the $jth$ input dimension. The center value of the second-order membership function was the output value of the first-order membership function; the formula was as follows:

$$
\mu_{ij} = e^{-\left(\frac{x_j - m_{ij}}{\sigma_{ij}}\right)^2}
\tag{32}
$$

To obtain the firing strength of the type-II fuzzy rule, the $\alpha$ cut was used for cutting the first type of fuzzy set. The $\alpha$ cut formula was expressed as:

$$
\widetilde{A}_\alpha(x) = \left\{ (x, u_1) \,\middle|\, u_{\underset{A}{\sim}}(x, u_1) \geq \alpha \right\}
\tag{33}
$$

where $u_1$ is the first-order membership function value. We set five $\alpha$ cut values—$\alpha = \{1, 0.8, 0.6, 0.4, 0.2\}$—and defined $\tilde{A}_{ij}$ as the second type of fuzzy set, which was expressed as:

$$
\widetilde{Z}_{ij}(x_i) = \bigcup_{i=1}^{5} \alpha_i \cdot \left[ l_{ij\alpha_i}, r_{ij\alpha_i} \right], i = 1, \cdots 9, j = 1, 2
\tag{34}
$$

where $\left[ l_{ij\alpha_i}, r_{ij\alpha_i} \right]$ was the cut set interval of the second-order membership function, $l_{ij\alpha_i}$ was the lower bound, and $r_{ij\alpha_i}$ was the upper bound.

Different from a traditional fuzzy, a type-II fuzzy inference engine deals with the interval input. The firing strength of the $j^{th}$ rule could be obtained from the following equation:

$$
f_j = U_{i=1}^{5} \alpha_i \left[ \prod_{i=1}^{2} l_{ija_i}, \prod_{i=1}^{2} r_{ija_i} \right] = U_{i=1}^{5} \alpha \left[ vl_{ja_i}, vu_{ja_i} \right], j = 1, 2, \ldots 9
\tag{35}
$$

where $vl_{j\alpha_i}$ and $vu_{j\alpha_i}$ were the pure quantities that represented the lower and upper limits of the firing strength of the $jth$ fuzzy rule in the $\alpha$ cut set.

In the next step, the type reducer function was used to reduce the output strength of the inference engine. The Karnik–Mendel (KM) algorithm was used to reduce the order of interval of the second type of fuzzy inference [25,26]. The calculated formula was:

$$
out = \bigcup_{i=1}^{5} \alpha_i \cdot \left[ \underset{-\alpha_i}{o}, \overline{o}_{\alpha_i} \right]
\tag{36}
$$

where $\underset{-\alpha_i}{o}, \overline{o}_{\alpha_i}$ represented the output of the lower and upper limits of the KM algorithm, which could be calculated by using Equations (22) and (23).

$$
\underset{-\alpha}{o} = \frac{\sum_{k=1}^{L} c_k o_{k\alpha}^{-} + \sum_{k=L+1}^{9} c_k \underset{-k\alpha}{o}}{\sum_{k=1}^{L} o_{k\alpha}^{-} + \sum_{k=L+1}^{9} \underset{-k\alpha}{o}}
\tag{37}
$$

$$o_\alpha^- = \frac{\sum\limits_{k=1}^{R} c_k o_{-k\alpha} + \sum\limits_{k=R+1}^{9} c_k o_{k\alpha}^-}{\sum\limits_{k=1}^{R} o_{-k\alpha} + \sum\limits_{k=R+1}^{9} o_{k\alpha}^-} \tag{38}$$

where $L$ and $R$ were the values obtained from the KM algorithm iterations and $o_{k\alpha}^-$ and $o_{-k\alpha}$ were the results sorted from small to large of $vl_{j\alpha_i}$ and $vu_{j\alpha_i}$, which were based on $c_k$. This was then one part of the fuzzy rules. The output of the defuzzifier was:

$$\hat{y} = \frac{\sum\limits_{\alpha} \alpha \frac{o_{-\alpha} + o_\alpha^-}{2}}{\sum\limits_{\alpha} \alpha} \tag{39}$$

## 6. Experimental Results

The four experiments of the purity control of the SMB system using the type-II controller are presented in this section. Figures 7–10 show the separation effects of materials A and B at the outlets of the extraction and raffinate.

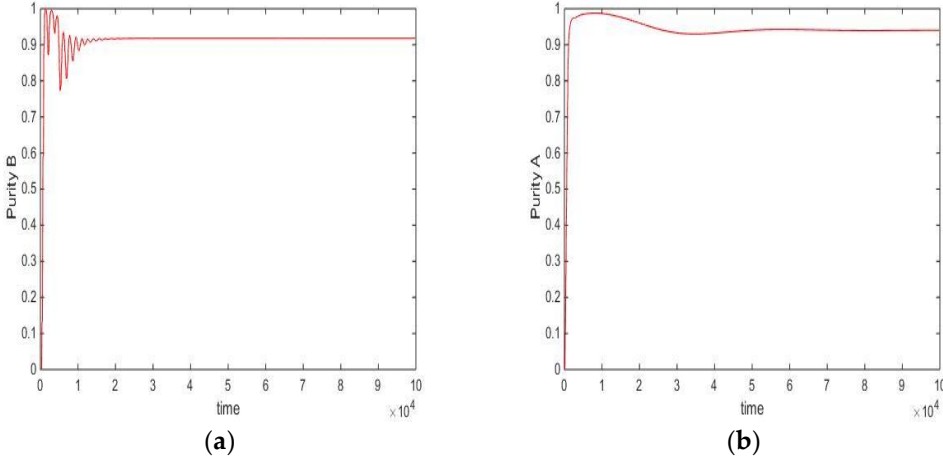

(a)                                                              (b)

**Figure 7.** Control results of the first experiment (switch time = 180 s; desired A = 94%, desired B = 92%). (**a**) Extraction outlet (material B); (**b**) raffinate outlet (material A).

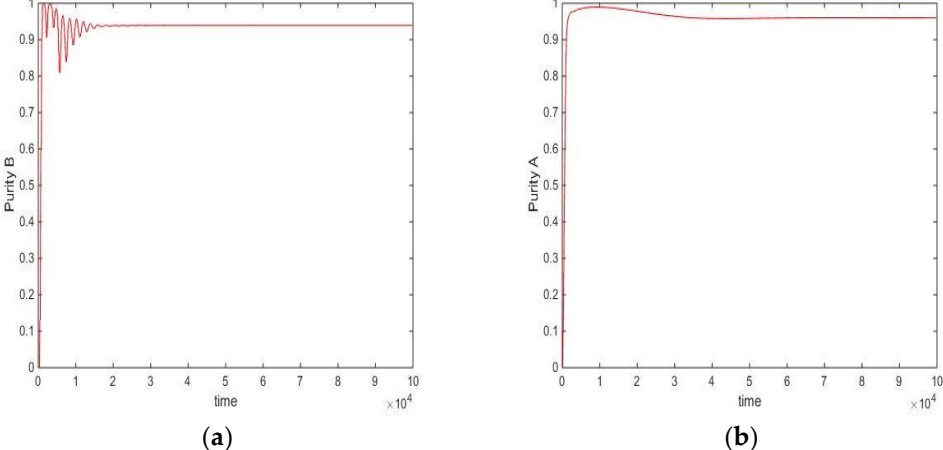

(a)                                                              (b)

**Figure 8.** Control results of the second experiment (switch time = 180 s; desired A = 96%, desired B = 94%). (**a**) Extraction outlet (material B); (**b**) raffinate outlet (material A).

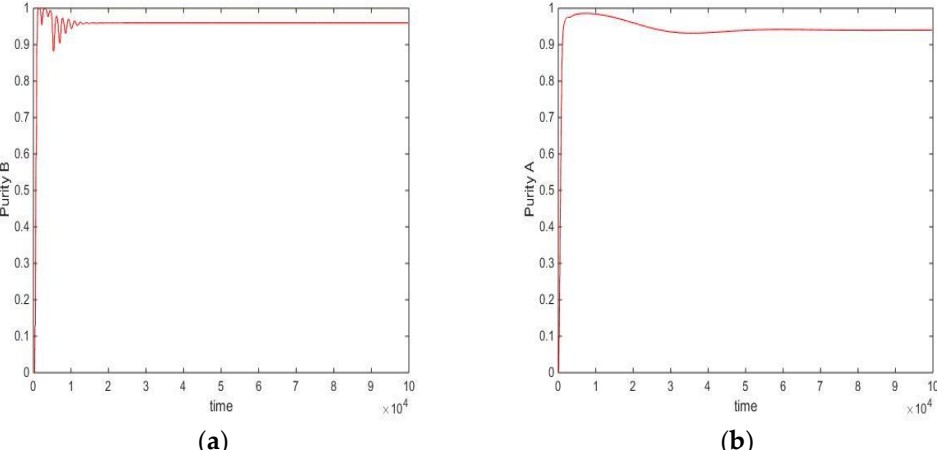

**Figure 9.** Control results of the third experiment (switch time = 180 s; desired A = 94%, desired B = 96%). (**a**) Extraction outlet (material B); (**b**) raffinate outlet (material A).

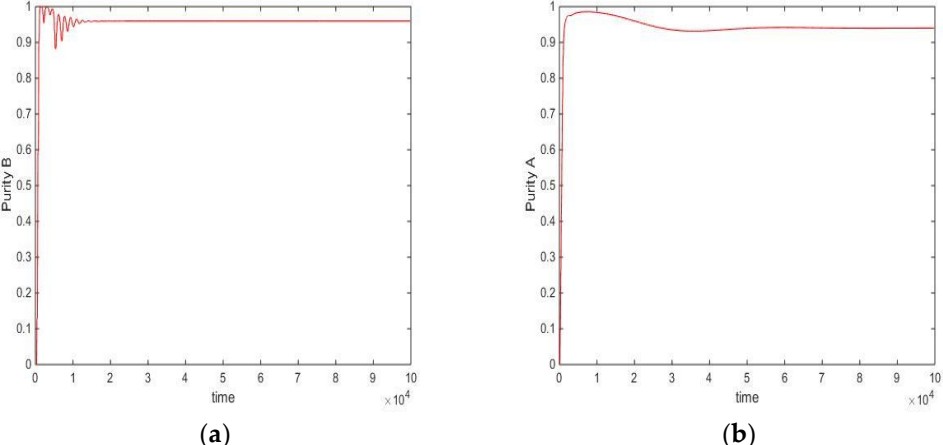

**Figure 10.** Control results of the fourth experiment (switch time = 180 s; desired A = 93%, desired B = 95%). (**a**) Extraction outlet (material B); (**b**) raffinate outlet (material A).

In the first experiment, the switching time of the SMB was set to be 180 s. The desired control purity of material A was 94% and the purity of material B was 92%. Figure 7a,b present the control results of the type-II fuzzy controller. The actual control purity of A was 94.04% and the purity of B was 91.89%. In the second experiment, the switching time of the SMB was still 180 s, but the desired control purity of material A was 96% and the purity of material B was 94%. Figure 8a,b present the control results; the actual control purity of A was 96% and that of material B was 93.9%. In the third experiment, the switching time of the SMB was still 180 s, but the desired control purity of material A was 94% and the purity of material B was 96%. The control results are shown in Figure 9a,b. The actual control purity of A was 94%; that of material B was 95.97%. In the fourth experiment, the switching time of the SMB was set to be 178 s; the desired control purities of material A and material B were 93% and 95%, respectively. The control results are shown in Figure 10a,b. The actual control purity of A was 93%; that of material B was 94.95%.

In this study, a traditional fuzzy controller and PID controller were also implemented for the SMB control as a comparison. Figure 11 shows the performances from the traditional fuzzy controller, the PID controller, and the type-II fuzzy controller. In this implementation, the switch time was 180 s; the desired control purity of material A was 93% and the purity of material B was 95%. Figure 12 shows the control results of the three controllers for another implementation. In this implementation, the switch time was 178 s; the desired control purity of material A was 96% and the purity of material B was 94%. In this implementation,

the switch time was 178 s; the desired control purity of material A was 94% and the purity of material B was 96%.

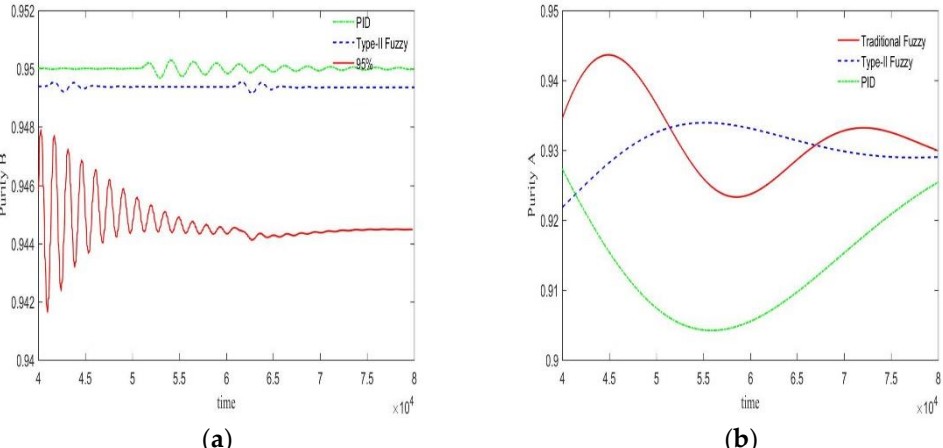

**Figure 11.** The control results of the three controllers (switch time = 180 s; desired A = 93%, desired B = 95%). (**a**) Extraction outlet (material B); (**b**) raffinate outlet (material A).

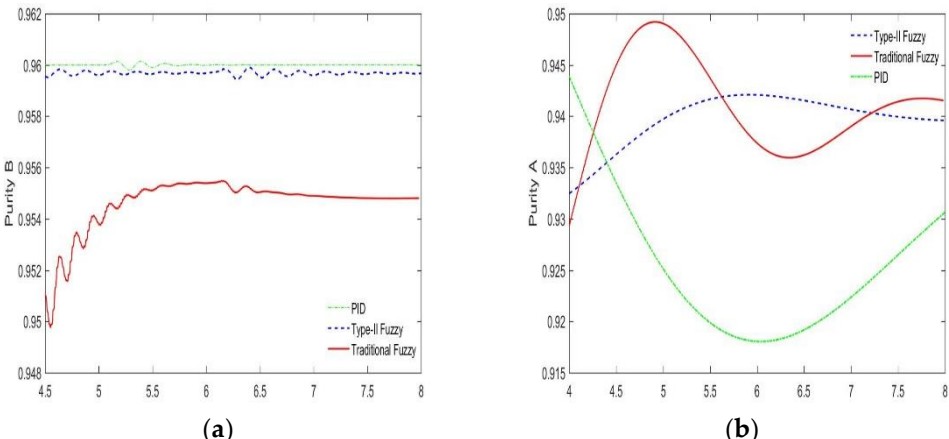

**Figure 12.** The control results of the three controllers (switch time = 180 s; desired A = 93%, desired B = 95%). (**a**) Extraction outlet (material B); (**b**) raffinate outlet (material A).

From the results shown in Figures 11–13, it is clearly shown that there were steady-state errors and fluctuations in the purity of the extraction and raffinate materials controlled by the traditional fuzzy controller. However, the type-II controller had no steady-state errors and could maintain a steady state. For the PID controller, the control of the extraction outlet was relatively stable, but the purity of the raffinate outlet fluctuated too much.

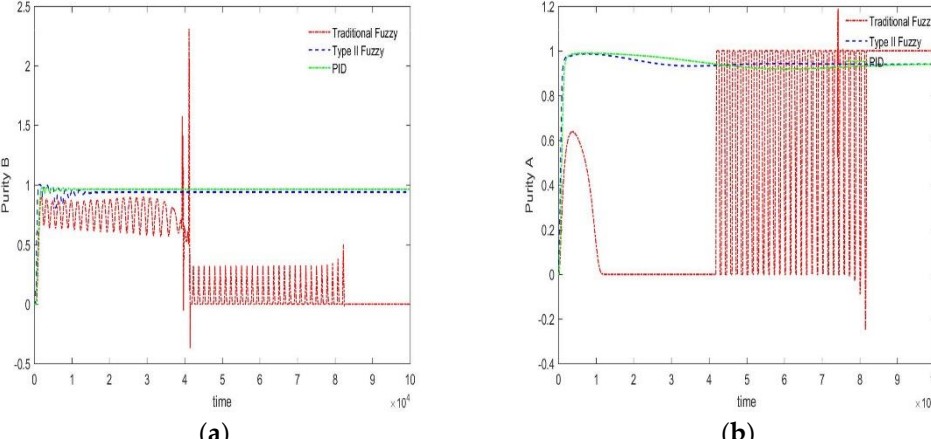

**Figure 13.** The control results of the three controllers (switch time = 178 s; desired A = 94%, desired B = 96%). (**a**) Extraction outlet (material B); (**b**) raffinate outlet (material A).

In Figure 13, when the switching time parameter was 178 s, it can be seen that the traditional fuzzy controller had ill-conditioned characteristics whereas the control effect of the type-II controller was still robust. Surprisingly, the PID controller was also relatively stable.

We then explored the stability of the three controllers. The control effects of the three controllers on the changes in the adsorbent parameters, feed concentration, and switching time were implemented and observed. Figures 14–19 show the separation results controlled by the three controllers. Each figure presents the results of five purity control experiments. The desired purity control of material A was 93%, 94%, 95%, 96%, and 97%; the desired purity control of material B was 92%, 93%, 94%, 95%, and 96%.

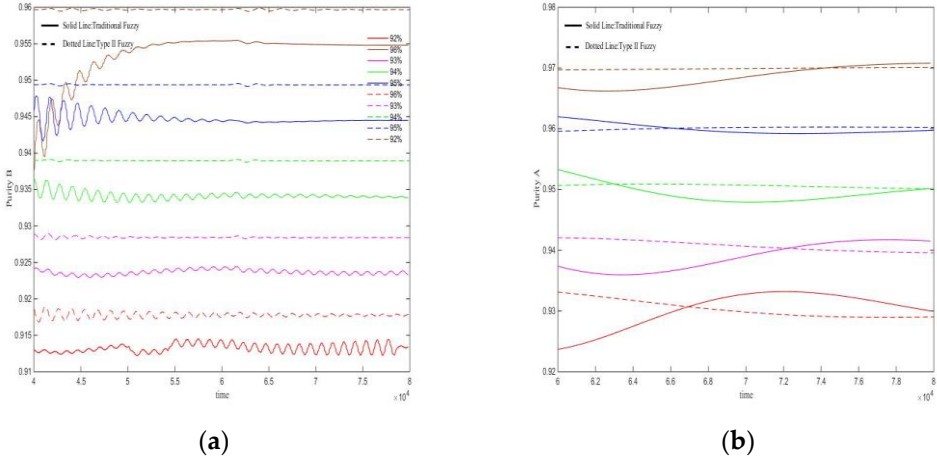

**Figure 14.** Under the change of the adsorbent parameter $H_A = 0.01 \rightarrow 0.03$. (**a**) Extraction outlet (material B); (**b**) raffinate outlet (material A).

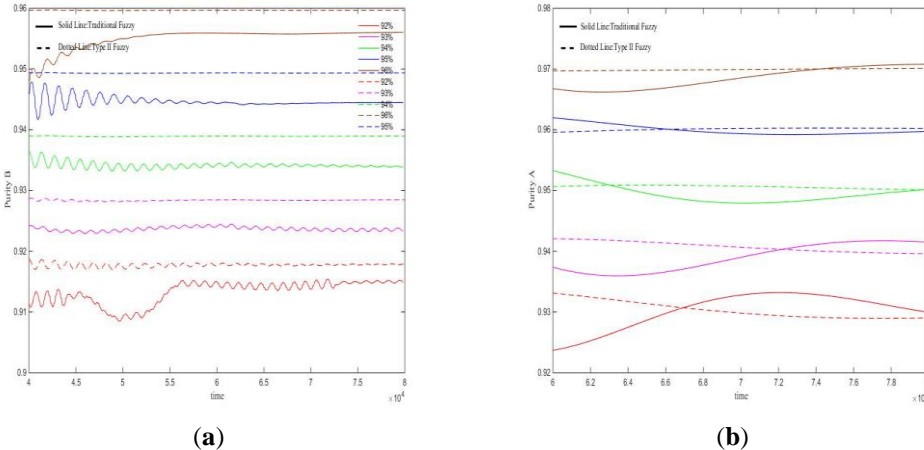

**Figure 15.** Under the change of the feed port concentration $C_f = 4.5 \rightarrow 5.2$. (**a**) Extraction outlet (material B); (**b**) raffinate outlet (material A).

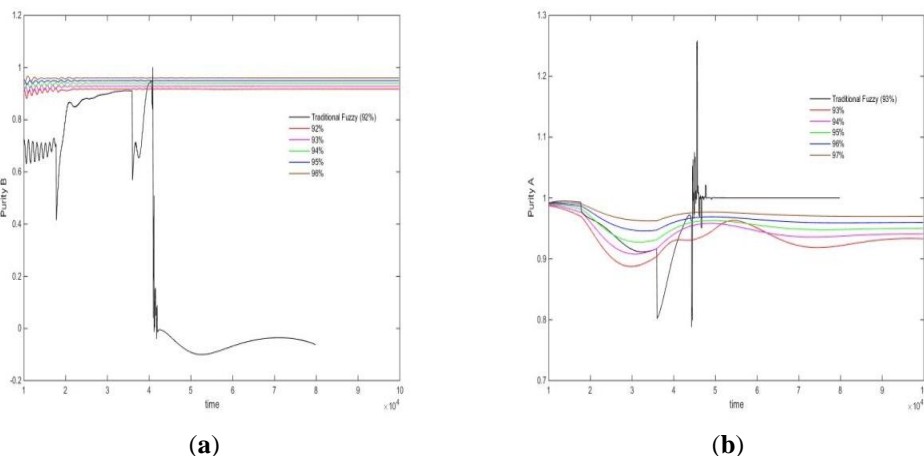

**Figure 16.** Under the change of the switch time $\theta = 178s \rightarrow 182s$. (**a**) Extraction outlet (material B); (**b**) raffinate outlet (material A).

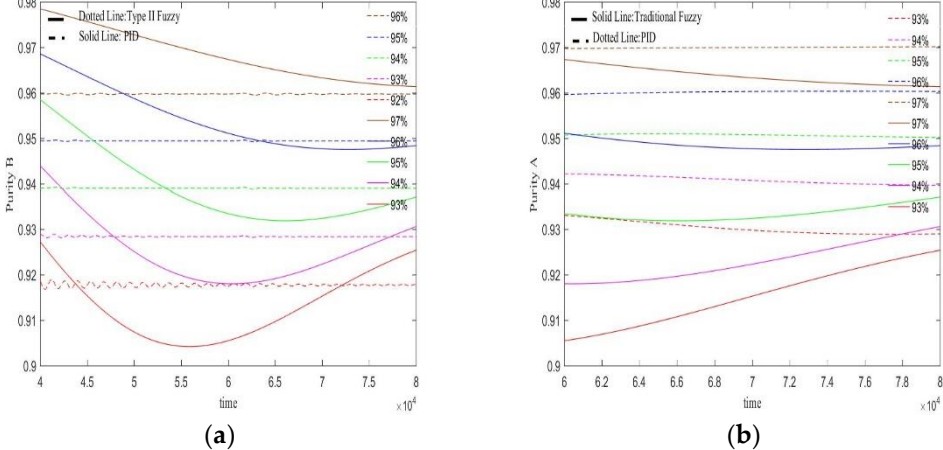

**Figure 17.** Under the change of the adsorbent parameter $H_A = 0.01 \rightarrow 0.03$. (**a**) Extraction outlet (material B); (**b**) raffinate outlet (material A).

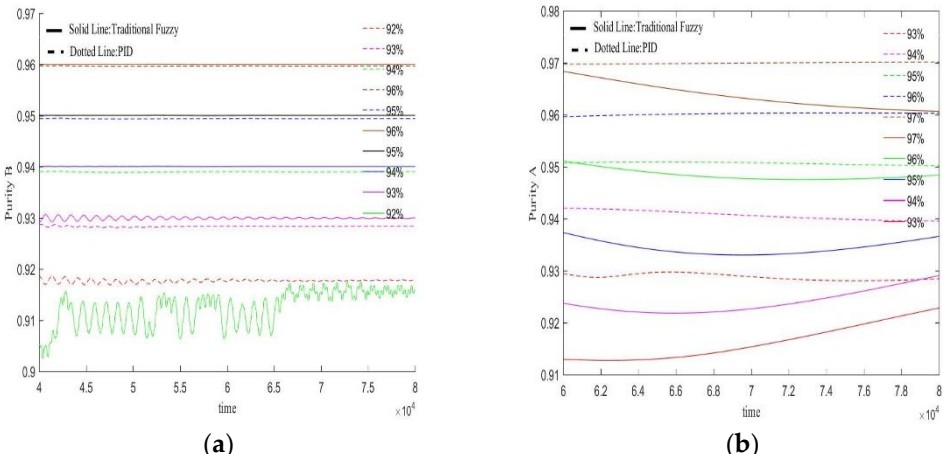

(a)

(b)

**Figure 18.** Under the change of the adsorbent parameter $C_f = 4.5 \rightarrow 5.2$. (**a**) Extraction outlet (material B); (**b**) raffinate outlet (material A).

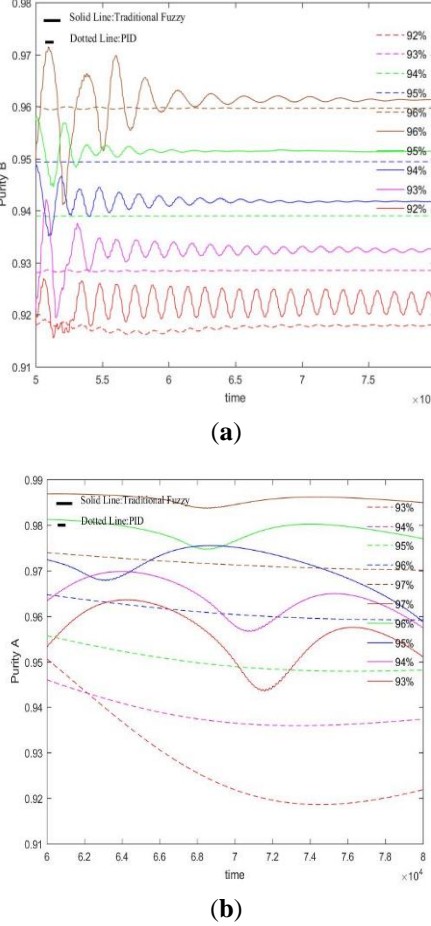

(a)

(b)

**Figure 19.** Under the change of the switch time $\theta = 178s \rightarrow 182s$. (**a**) Extraction outlet (material B); (**b**) raffinate outlet (material A).

In the case of changing the adsorbent parameters ($H_A$), it can be seen from Figure 14 that the traditional fuzzy controller not only had the problem of steady-state errors in the purity control of material B, but also had an oscillation phenomenon. However, the control effect of the type-II controller remained stable. For the purity of material A, the fluctuation of the traditional fuzzy controller was obviously larger than that of the type-II fuzzy controller.

In the case of changing the feed port concentration ($C_f$), the experimental results in Figure 15 show that the control performance of two controllers was similar to that in Figure 13.

In Figure 16, it can be observed that, under the change of switching time, the traditional fuzzy controller had ill-conditioned characteristics, but the type-II fuzzy controller was still very stable.

From the experimental results shown above, the type-II controller was very accurate in controlling the purity of the two outlets, with almost no steady-state errors. For the changes in the system parameters such as the adsorbent parameters, feed port concentration, and switching time, the type-II controller had excellent robustness and adaptability.

Finally, we compared the disturbance of the PID and type-II controllers. Figures 17–19 show the separation results. Although the PID controller did not show any pathological characteristics under the disturbance of the adsorbent, feed inlet, and switching time parameters, the purity of the extraction port and raffinate port fluctuated too much and there was a relatively large amplitude of flutter, especially when the concentration of the feed inlet and switching time changed and the purity of the extraction port significantly vibrated. Type-II obviously showed good robustness and adaptability, which were the biggest advantages that the PID controller did not have.

## 7. Conclusions

In this paper, a type-II fuzzy controller in an SMB system control was presented. As mentioned above, the SMB is a chromatographic separation technique that is widely used in the manufacture of chemical and biopharmaceutical products. The SMB is a very complex system and there are too many parameters involved in its control process. If the optimized parameters are obtained through trial-and-error methods to control the SMB separation process, the cost is certainly very high and unfeasible. Even if the initial control parameters of the SMB are set based on a technician with full experience, the SMB control still cannot reach the best condition.

In our study, the type-II controller was very accurate in controlling the purity of the two outlets, with almost no steady-state errors. The system was robust and adaptable to changes in parameters such as the adsorbent, feed port concentration, and switching time parameters. Compared with the traditional fuzzy controller and PID controller, the type-II fuzzy controller was not only more accurate in the control, but also could reach a steady state in a shorter control time. Its robustness and adaptability to various parameter changes and disturbances were much superior to the traditional fuzzy controller.

Although the type-II fuzzy controller had a stronger fault tolerance, the model also had force parameters and mean and variance parameter matrices, which were bound to affect the final application effect. Therefore, if the learning ability could be improved on the basis of this to make it more adaptive and robust, it is undoubtedly of great significance for the control of SMB systems. The neural network has a super nonlinear learning ability. Building a type-II fuzzy neural network control method with an adaptive learning ability has great theoretical significance in promoting nontraditional affine nonlinear control systems and has a great application value in chemical process controls.

**Author Contributions:** Conceptualization, R.-C.H. and C.-F.X.; methodology, C.-F.X.; software, C.-F.X.; validation, R.-C.H. and C.-F.X.; formal analysis, R.-C.H.; resources, R.-C.H.; data curation, C.-F.X.; writing—original draft preparation, C.-F.X.; writing—review and editing, R.-C.H.; visualization, C.-F.X.; supervision, R.-C.H.; project administration, R.-C.H. All authors have read and agreed to the published version of the manuscript.

**Funding:** This research was supported in part by the National Natural Science Foundation of China under Grant 62071123, 61601125, by the Fujian Province Education Hall Youth Project (number: JAT170679), by the Fujian Natural Science Foundation Project (number: 2019J01887), by the Fujian Provincial Marine Economic Development Subsidy Fund Project (number: FJHJF-L-2019-7), by the Electronic Information and Engineering Institute of the Fujian Normal University, and by the

School of Economics of Fujian Normal University, Key Laboratory of Nondestructive Testing, Fujian Polytechnic Normal University.

**Institutional Review Board Statement:** Not applicable.

**Informed Consent Statement:** Not applicable.

**Data Availability Statement:** No other institutional data is involved, which is realized through model simulation.

**Conflicts of Interest:** The authors declare no conflict of interest.

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
