# Peer review of "Purity Control Based on a Type-II Fuzzy Controller for a Simulated Moving Bed"

_processes, doi:10.3390/pr10112437_

Round 1

Author Response

 Thank the reviewers for the precious suggestions, according to your requirements, introduce some changes as follow:

  1. We add the parts: (1)So far, the vast majority of the industry still uses relatively simple PLC control, and some use specific controllers based on materials to be separated, such as model predictive control and PID controller. Therefore, there is no unified intelligent processing mode. (2) Type-II fuzzy control is based on type-II fuzzy sets, which makes the type-II fuzzy controller have more flexible attribution function values.(3)The results show that the type-II fuzzy controller is not only more accurate in control, but also better in robustness and adaptability than the ordinary fuzzy controller and PID controller. The three parts corresponding to 2, 3 and 4 in the abstract question proposed by the reviewer.
  2. An introduction should clearly highlight the motivation, problem statement, the objective of the paper, gap in the existing research and the novelty of the conducted research. To solve this problem, we have added the following description in the introduction section: 

    (1)Generally, these studies are for specific equipment and separation materials, not the general solution. For example, sugar separation in the food industry may not be applicable to the chemical industry. In terms of calculation model, most of them are based on finite element calculation method. Although the precision of finite element calculation is high, the calculation efficiency is not high enough, which is not suitable for the application of real-time online control ,in this paper, it used finite difference method to enhance calculation efficiency. The robustness and adaptability of the controller used are lack of corresponding research. With the change of environmental parameters, the results are easy to lead to the failure of the separation effect. In the practical application of SMB, it is often necessary to adjust according to the changes of equipment and environment, so a more intelligent automatic control technology is needed. Compared with traditional nonlinear control methods such as feedback linearization, synovium, pole placement, etc., fuzzy control based on fuzzy mathematics does not need too much mechanism model knowledge, so the use of fuzzy control technology is extremely convenient and effective in mathematical processing, especially for nonlinear systems with complex structures.

     (2)In the face of highly sensitive SMB nonlinear systems, the simple use of fuzzy control technology can not guarantee the robustness and adaptability of its control. The fuzzy controller depends on the fuzzy application rules and the force size and other parameters. Similarly, if the controller parameters are not properly selected, it is easy to cause it to cross the feasible separation area of the SMB system in the control, resulting in pathological characteristics. In this paper, we propose a more effective type-II fuzzy controller. Type-II fuzzy control is based on type-II fuzzy sets, which makes the type-II fuzzy controller have more flexible attribution function values, because the value of type-II fuzzy function is also a fuzzy set. So it takes the design of type-II fuzzy inference more flexible, and the system's error tolerance also increases, furthermore, the robustness and adaptability of the system control are also enhanced. The architecture of this paper is to design a controller for the type-II fuzzy controller, control the output purity of the SMB system, observe the difference in control accuracy compared with traditional fuzzy control and PID control, and conduct disturbance resistance experiments, mainly involving the three most important parameters of the SMB system, such as switching time, concentration of the feed inlet and the adsorption capacity of the adsorbent. Finally, compare the differences in robustness and adaptability of the controller. 
  3. Add more theoretical and mathematical description about the problem。In response to the comments of reviewers, we have added a large part to describe the discretization process of the dynamic system of the entire simulated moving bed. Since too many formulas are involved, see the revised version for details。
  4. The paper has supplemented the relevant situation of using PID controller for control, which is shown in figures 11,12,13, 17,18,19.
  5. Cite latest work of high rank journals and remove low rank journals。

    We are very sorry for this problem. After checking the relevant literature, there are really few literatures on model analysis. There are indeed many literatures on material analysis in the chemical industry, but they are not too relevant to the research direction of this paper. I feel sorry again.

    Finally, thank you very much for your constructive comments, especially for more relevant experiments on the mathematical details of the model. Thank you.

Reviewer 2 Report

This paper presented a type-II fuzzy controller in SMB system control. This research is interesting but there are concerns related to the manuscript of this research. The authors should be addressing the following concerns in the first round:

1. I encourage you to add more detail about your core contributions in the abstract. Abstract has five-section and you should follow the best practices in your area! Please also mention the novelties and results in the abstract.

2. The keywords are not standard.

3. The Introduction section is not well structured. The authors should describe the main contributions of the research in the Introduction section. It should be clearly defined what is the contribution of this research.

4. In the last paragraph of the Introduction section, the authors must describe the rest of the paper.

5. Please add research gap section.

6. There are a lot of line spaces between Table 1 and its title. Please check and correct it.

7. Check the English presentation of this paper and improve some sentences.

8. Findings, limitations, and recommendations of this paper can be discussed more in the conclusion section.

9. Please bring and focus on future research directions.

Author Response

We gratefully appreciate for your valuable suggestion, according to your comments, the revised part is as follows:

1.We add the novelties and results in the abstract which were marked by red, the add part were as follow:

    So far, the vast majority of the industry still uses relatively simple PLC control, and some use specific controllers based on materials to be separated, such as model predictive control and PID controller. Therefore, there is no unified intelligent processing mode.

    The results show that the type-II fuzzy controller is not only more accurate in control, but also better in robustness and adaptability than the ordinary fuzzy controller and PID controller.

2.The keywords have been modified as: Simulated Moving Bed ; Type-II Fuzzy Controller; PID Controller 

3. The contribution of this research as follow:In the face of highly sensitive SMB nonlinear systems, the simple use of fuzzy control technology can not guarantee the robustness and adaptability of its control. The fuzzy controller depends on the fuzzy application rules and the force size and other parameters. Similarly, if the controller parameters are not properly selected, it is easy to cause it to cross the feasible separation area of the SMB system in the control, resulting in pathological characteristics. In this paper, we propose a more effective type-II fuzzy controller. Type-II fuzzy control is based on type-II fuzzy sets, which makes the type-II fuzzy controller have more flexible attribution function values, because the value of type-II fuzzy function is also a fuzzy set. So it takes the design of type-II fuzzy inference more flexible, and the system's error tolerance also increases, furthermore, the robustness and adaptability of the system control are also enhanced. The architecture of this paper is to design a controller for the type-II fuzzy controller, control the output purity of the SMB system, observe the difference in control accuracy compared with traditional fuzzy control and PID control, and conduct disturbance resistance experiments, mainly involving the three most important parameters of the SMB system, such as switching time, concentration of the feed inlet and the adsorption capacity of the adsorbent. Finally, compare the differences in robustness and adaptability of the controller.

4. We describe the rest of the paper which in the last paragraph of the Introduction section: Other studies using mixed multiple models are described below. Suvarov et al. used self-adjusting control to adjust the spatial position of adsorption and desorption waves, and then adjusted the purity and productivity of extraction liquid and extraction flow. Such predictive control technology has been widely used in program control [17]. Song et al. put forward a new operation strategy called Simcon. This method improves the separation performance of SMB chromatography by controlling the outlet and inlet simultaneously. This control operation is simple, but the accuracy is not too high[18]. Carols et al. propose a new approach which is based on the combination of the wave theory and Multi-Model Predictive Control (MMPC). The wave theory provides the theoretical framework[19]. Based on the mathematical model, a numerical solution process is proposed to simulate the transient and steady state of the moving bed by Leao et al. It is also a theoretical architecture and computing model[20]. In the simulation study, Ju WenLee proposed a simplified process model with linear isotherms to estimate the process state of SMB chromatography. It find the optimal operating parameter conditions through the "switch by switch" switching operation within the moderate nonlinear range of Langmuir isotherm [21]. Yang et al. proposed an optimization strategy based on the improved moving asymptote algorithm. Research shows that the optimal controller based on the improved moving asymptote method can dynamically control and optimize the process of the simulated moving bed[22].

5. We add research gap section:

Generally, these studies are for specific equipment and separation materials, not the general solution. For example, sugar separation in the food industry may not be applicable to the chemical industry. In terms of calculation model, most of them are based on finite element calculation method. Although the precision of finite element calculation is high, the calculation efficiency is not high enough, which is not suitable for the application of real-time online control ,in this paper, it used finite difference method to enhance calculation efficiency. The robustness and adaptability of the controller used are lack of corresponding research. With the change of environmental parameters, the results are easy to lead to the failure of the separation effect. In the practical application of SMB, it is often necessary to adjust according to the changes of equipment and environment, so a more intelligent automatic control technology is needed. Compared with traditional nonlinear control methods such as feedback linearization, synovium, pole placement, etc., fuzzy control based on fuzzy mathematics does not need too much mechanism model knowledge, so the use of fuzzy control technology is extremely convenient and effective in mathematical processing, especially for nonlinear systems with complex structures.

6.There are a lot of line spaces between Table 1 and its title. We have check and correct it.

7. We have check the English presentation of this paper and improve some sentences which were marked by red.

8. Limitations, and recommendations of this paper were discussed in conclusion as follow: Although the Type II fuzzy controller has stronger fault tolerance, the model also has force parameters and mean and variance parameter matrices, which are bound to affect the final application effect. 

9. Future research directions are added in conclusion:  Therefore, if the learning ability can be increased on the basis of it to make it more adaptive and robust, it is undoubtedly of great significance for the control of SMB systems, The neural network just has super nonlinear learning ability. Building a type-II fuzzy neural network control method with adaptive learning ability has great theoretical significance for promoting non-traditional affine nonlinear control systems, and has great application value in chemical process control.

Reviewer 3 Report

The research is focused on the controller of the simulated moving bed. The use of fuzzy controller is the best choice for this highly complex nonlinear system. The author proposed to use the popular Type II fuzzy controller and compared with the traditional fuzzy controller to obtain more accurate results. When performing the disturbance analysis, the authors found that the Type II fuzzy controller has excellent robustness and adaptability, Type II fuzzy controller provides a good reference for industrial SMB material separation control.

At this stage, the SMB industry still adopts the relatively backward PLC controller. This manuscript provides a good research direction for the chromatographic separation industry to move towards intelligent industry.

Therefore, I suggest accepting this manuscript after some details are revised.

Some details:

1.       One suggestion, the background of simulated moving bed is well clarified. Some references from journals for example Processes may be referred.

2.       Show the full name of abbreviations when it appears first time, for example SMB. It appears in abstract, however, it might be mentioned in introduction part. In table 3 and table 4, the NB NS ZE PS PB may be explained in manuscript.

3.       The font of table 1 may be revised to set in a full page.

4.       The font of equation 10 might be consistent with other equations.

5.       Figure 2 to 4, and 6 are not clear. The symbol of “enter” symbol might be removed during Microsoft word processing.

6.       Double check the format of references. The quotation marks “”  are not required. The font may also be revised to by Palatino Linotype not times new roman.

Author Response

Thanks for the reviewer positive comments.

  1.We went to the Processes journal to look for relevant literature, and found that it was not related to my research direction.

2.We revised the first line of the summary as follows: The control of simulated moving bed abbreviated form of SMB s always a challenging chemical control topic due to its complexity and nonlinearity .

3. We have revised the table 1, and let table name and table content in the same page and relocated it.

4. Thank you for your careful discovery,we have revised equation 10 ,let it consistent with other equations.

5.We reprocessed pictures 2, 4 and 6 to make them look more professional。

6.We rechecked the references section to make the font conform to the format of the journal. Thank you again.

Round 2

Reviewer 1 Report

The revised manuscript can be considered for publication.

Author Response

Thank the reviewer for your comments, which are very helpful to my research. I add several rows to describe  the paper architecture. Thank you again.

1.The English language was rechecked and some spelling and grammar errors were corrected. The modified parts were marked with red and yellow background

Reviewer 2 Report

The authors have responded to most of the concerns raised and made corrections to the manuscript. But, one of the comments was wrongly applied in the manuscript.

Authors should get help from previously published articles on how to write descriptions related to the "rest of the paper". Because the last paragraph of the Introduction section in articles is written in almost the same style.

To help the authors, I added sample text below for authors to draw inspiration from that.

"The rest of the paper is organized as follows. Section 2 presents the mathematical model of the SMB. In Section 3, the Crank-Nicolson method to numerate PDEs is presented. Section 4 presents a ... "

Please, add similar style sentences for the sections 4 to 6, so that the readers of the article will understand what sections are discussed in the following.

Author Response

Thank the author for his comments, which are very helpful to my research.

1.According to the requirements of the reviewers, descriptions of the rest of the paper are added, and these parts are marked with green. Thank you again!
